

# Comparative effects of square-stepping and strengthening exercises on cognitive and balance functions in chronic obstructive pulmonary disease: a randomized clinical trial

Alp Özel[1], Eylem Tütün Yümin[1] and Suat Konuk[2]

[1] Department of Physiotherapy and Rehabilitation/Faculty of Health Sciences, Bolu Abant Izzet Baysal University, Bolu, Turkey
[2] Department of Chest Diseases/Faculty of Medicine, Bolu Abant Izzet Baysal University, Bolu, Turkey

Corresponding author
Alp Özel, alpozel@ibu.edu.tr

## ABSTRACT

**Background**. Cognitive impairment and balance dysfunction are common in individuals with chronic obstructive pulmonary disease (COPD), yet targeted interventions remain limited. Square-stepping exercise (SSE), a structured multitasking intervention involving progressive, multi-directional step patterns, combines cognitive and motor challenges. This study aimed to compare the effects of SSE and traditional strengthening exercises (SE) on cognitive function and balance in individuals with COPD through a telerehabilitation model.

**Methods**. This randomized clinical trial included 34 male individuals with mild to moderate COPD (mean age: $63.91 \pm 6.98$ years), randomly assigned to SSE and SE groups ($n = 17$ each). Both groups participated in supervised telerehabilitation sessions three times per week for eight weeks. Cognitive function was assessed using the Montreal Cognitive Assessment (MoCA; 0–30 points, with higher scores indicating better cognition), while the Standardized Mini-Mental State Examination (SMMSE; 0–30, cutoff $< 23$) was used as a screening tool to exclude significant cognitive impairment. Balance performance was evaluated using the Biodex Balance System, including the overall stability index, anterior/posterior index, and medial/lateral index (lower scores indicate better balance). Perceived breathlessness (dyspnea) was assessed with the Modified Medical Research Council (mMRC) scale (0–4), and disease impact with the COPD Assessment Test (CAT; 0–40, $\geq 10$ indicating high symptom burden). Comorbidity severity was evaluated using the modified Charlson Comorbidity Index (CCI; higher scores indicate greater severity). Data normality was assessed using the Shapiro–Wilk test. Independent sample t-tests were used for parametric between-group comparisons, and Mann–Whitney $U$ tests were applied for non-parametric data. Paired sample $t$-tests and Wilcoxon signed-rank tests were used for within-group comparisons. The level of statistical significance was set at $p < 0.05$.

**Results**. Both groups showed significant within-group improvement in MoCA scores ($p = 0.01$ for both). However, the SSE group demonstrated greater improvements in balance parameters, particularly in the overall stability index ($p = 0.014$) and anterior/posterior stability index ($p = 0.05$), compared to the SE group. The SE group showed limited improvements, primarily in static balance conditions ($p = 0.029$).

Although cognitive gains were similar between the groups, balance improvements were more pronounced in the SSE group.

**Conclusions**. While both exercise modalities improved cognitive function in individuals with COPD, SSE led to superior outcomes in balance control. The multitasking design of SSE, requiring simultaneous cognitive processing and motor coordination, may underlie its enhanced impact on postural stability. These findings support SSE as a technically advantageous and accessible intervention in telerehabilitation for individuals with COPD.

# INTRODUCTION

Chronic obstructive pulmonary disease (COPD) is a heterogeneous, preventable condition marked by persistent respiratory symptoms and systemic effects (*Agusti et al., 2023*). Primary symptoms include breathlessness, coughing, and sputum production (*Jacobson, Lind & Persson, 2023*). Individuals with COPD frequently experience comorbid conditions, with cognitive impairment being a common yet understudied issue (*Siraj, 2023*). Studies show that cognitive impairment prevalence in individuals with COPD ranges from 10% to 77% (*Stellefson, Wang & Campbell, 2024*; *Campman & Sitskoorn, 2013*). Increasing physical activity may positively impact cognitive functions in these individuals (*Higbee & Dodd, 2021*).

According to the American Thoracic Society (ATS) and the European Respiratory Society (ERS), comprehensive rehabilitation programs including aerobic and muscle strengthening exercises (SE), along with patient education, are recommended for COPD management (*Holland et al., 2021*). However, alternative approaches, such as square-stepping exercise (SSE), are gaining interest due to their structured and cost-effective nature. Several factors contribute to the growing interest in SSE. Its systematic nature stems from its structured progression, where participants follow predefined step patterns that increase in complexity over time, allowing for gradual adaptation and improvement in both cognitive and physical functions. Additionally, SSE is regarded as a cost-effective intervention since it requires minimal equipment (typically just a marked mat) and can be performed in various settings, including home-based and telerehabilitation programs, thereby reducing the need for costly facilities or specialized supervision. While its accessibility and feasibility make it a practical option for rehabilitation, further research is needed to substantiate its cost-effectiveness compared to traditional exercise modalities (*Agusti et al., 2023*). SSE, though rarely used in respiratory diseases, has shown promise in addressing balance, muscle weakness, and cognitive impairments, particularly in older adults (*Yuliadarwati & Setiawan, 2023*).

Studies suggest that SSE improves dynamic balance while engaging cognitive functions. Research has demonstrated its benefits in older adults and elderly individuals, where it has been shown to enhance both cognitive and motor skills (*Muslimaini, Mirawati &*

*Mutnawasitoh, 2023*; *Bhanusali et al., 2016*). Its low technological requirements make it suitable for rehabilitation, offering benefits for individuals with COPD by addressing balance and cognition together. SSE is a multitasking program that integrates cognitive functions such as concentration, memory, and motor skills with physical effort, and its effectiveness has been particularly noted in studies involving elderly populations (*Siqueira, Shigematsu & Sebastião, 2024*).

SE are well known for enhancing muscle strength and postural stability, though their impact on cognitive functions in individuals with COPD remains underexplored. *Vilaró et al. (2010)* highlighted that muscle dysfunction, particularly peripheral muscle weakness, is a key risk factor for hospital readmission in COPD, underscoring the importance of SE to improve muscle function and stability. SE that focus on peripheral muscles have been shown to improve not only muscle strength but also postural stability. For instance, scapulothoracic exercises have shown benefits on chest mobility and respiratory muscle strength in COPD, indicating potential impacts on functional capacity (*Thongchote, Chinwaro & Lapmanee, 2022*). Interest in the impact of peripheral muscle SE on cognitive functions in COPD is growing but remains under-researched. This relationship is crucial, as cognitive impairment in COPD can reduce physical activity levels, leading to a detrimental cycle (*Van Beers et al., 2018*). *Gore, Blackwood & Ziccardi (2023)* identified a link between cognitive function, balance, and gait speed in older adults with COPD, suggesting that enhancing physical strength and stability could improve cognitive outcomes. This highlights the potential of SE as a dual-purpose intervention for physical and cognitive health in this population.

This study aims to evaluate the effects of multitasking exercise and strengthening exercise on cognitive and balance functions in individuals with COPD. SE are widely recognized as the standard approach in COPD rehabilitation, primarily focusing on muscle strength, functional capacity, and postural stability. However, their effects on cognitive functions and dynamic balance remain underexplored. Multitasking exercises, such as SSE, have been shown to integrate cognitive and motor challenges, potentially offering additional benefits beyond traditional SE.

Given that individuals with COPD frequently experience both balance impairments and cognitive decline, it is crucial to investigate whether a multitasking approach like SSE provides superior functional outcomes compared to the standard SE. This study aims to compare SSE and SE to determine whether SSE offers additional advantages in improving both cognitive function and postural stability when delivered through telerehabilitation.

## Motivation and study contribution

Due to the high prevalence of both balance impairments and cognitive dysfunction in individuals with COPD, there is a growing need for integrated rehabilitation approaches that simultaneously target both domains. Traditional rehabilitation programs primarily focus on physical reconditioning and muscle strengthening, which improve postural control and functional capacity. However, they often fail to engage cognitive domains such as attention, memory, and executive function. These cognitive aspects are increasingly recognized as critical contributors to fall risk and disability in this population.

Furthermore, access to comprehensive rehabilitation services is frequently limited due to reduced mobility, geographic barriers, or healthcare system constraints. These challenges highlight the importance of developing low-cost, scalable, and remotely applicable interventions.

Multitasking-based programs such as SSE offer a promising alternative. They are designed to stimulate both cognitive and motor functions through structured and progressively challenging step patterns. Although SSE has shown benefits in older adults, its use in individuals with COPD, particularly in telerehabilitation settings, has been rarely studied. This study aims to address this gap by evaluating the feasibility and effectiveness of SSE compared to conventional strengthening exercises in a supervised, home-based telerehabilitation program. The main contributions of this study are as follows:

- It is the first randomized clinical trial to compare the effects of square-stepping exercise and traditional strengthening exercise on both cognitive function and balance in individuals with COPD.
- It investigates the feasibility and efficacy of supervised telerehabilitation, making the findings highly relevant for remote care models and digital health settings.
- It employs standardized and validated outcome measures, such as the Montreal Cognitive Assessment (MoCA) and Biodex Balance System indices, to provide a robust evaluation of both cognitive and postural outcomes.
- It addresses a literature gap by evaluating the cognitive benefits of a multitasking physical intervention (SSE) in a chronic respiratory disease population.
- It provides evidence supporting the use of multitasking-based exercises as a dual-purpose rehabilitation strategy for enhancing both cognitive health and balance control in COPD.

### Structure of the manuscript

The remainder of this manuscript is structured as follows: The Methods section outlines details on the study design, participant characteristics, intervention protocols, and assessment tools. The Results section presents the findings of the study, including both within- and between-group comparisons of cognitive and balance outcomes. The Discussion interprets these results in the context of existing literature, highlights clinical implications, and acknowledges the study's limitations. Finally, the Conclusion summarizes the key findings and offers suggestions for future research. This study included only male participants to ensure sample homogeneity and control for sex-related variability in cognitive and balance performance. However, this design choice may limit the generalizability of findings to the broader COPD population, particularly to female individuals.

## MATERIALS & METHODS

### Study design

This study was designed as a prospective, randomized, comparative clinical trial to ensure the robustness and reliability of our findings. Randomization serves as the gold standard for clinical trials because it minimizes selection bias, balances unknown confounding factors

across intervention groups, and enhances the generalizability of the results. Utilizing simple randomization *via* MedCalc 11.5.1 software ensured that each participant had an equal chance of being assigned to either the SSE or the SE group, thereby providing a solid foundation for comparing the effects of these interventions on cognitive function and balance in individuals with COPD.

Additionally, the adoption of a single blind method, where participants were unaware of their group assignment, aimed to reduce bias in participants' self-reporting and behavior. Although blinding the physiotherapists was not feasible, the study design minimized potential measurement bias by ensuring that the researchers analyzing the results were blind to group assignments. This methodological consideration strengthens the study's validity by protecting against observer bias.

A total of 38 individuals were randomly assigned in a 1:1 ratio into two groups using simple randomization *via* MedCalc 11.5.1 software: the SSE group ($n = 19$) and the SE group ($n = 19$). We conducted the study using a single blind method, where the participants were unaware of which group, they were assigned to. Although it was not feasible for the physiotherapist to be 'blind' to which patients received one or the other type of training, the assignment could have been hidden from the researchers analyzing the results. Implementing this approach in future studies may help reduce potential observer bias and further strengthen the validity of the findings.

## Participants

Male participants diagnosed with mild to moderate COPD were classified according to the GOLD 2021 ABCD system, which considers both the degree of airflow limitation and symptom burden. Airflow limitation was assessed using post-bronchodilator spirometry results, categorized as follows: Mild ($FEV_1 \geq 80\%$ predicted), moderate (50–79%), severe (30–49%), and very severe (<30%). Symptom severity was evaluated using the mMRC dyspnea scale and/or the COPD Assessment Test (CAT). This multidimensional classification ensured a comprehensive assessment of disease severity and informed participant eligibility for the study (*Global Initiative for Chronic Obstructive Lung Disease (GOLD), 2020*). At baseline, demographic and clinical data, including age, height, weight, and other relevant health parameters, were collected. However, as these variables were not the main focus of the study, they were not reassessed post-intervention. Participants were enrolled in the study at Bolu Abant Izzet Baysal University Training and Research Hospital, and the study was conducted between April 6, 2021, and February 15, 2022. The inclusion criteria included a confirmed COPD diagnosis according to GOLD 2021 guidelines. A Standardised Mini-Mental State Examination (SMMSE) score of 23 or higher was required for inclusion. In the Turkish population, a cutoff score of 23 has been established for the diagnosis of mild to moderate dementia (*Güngen et al., 2002*). This criterion was used to ensure that participants did not have significant cognitive impairment, allowing them to actively engage in the rehabilitation program. Participants were required to be between the ages of 50 and 80, have access to a smartphone and the internet, and be willing to adhere to the rehabilitation program. The exclusion criteria included continuous oxygen support, COPD exacerbation phase (defined as no drug change or antibiotic use due to

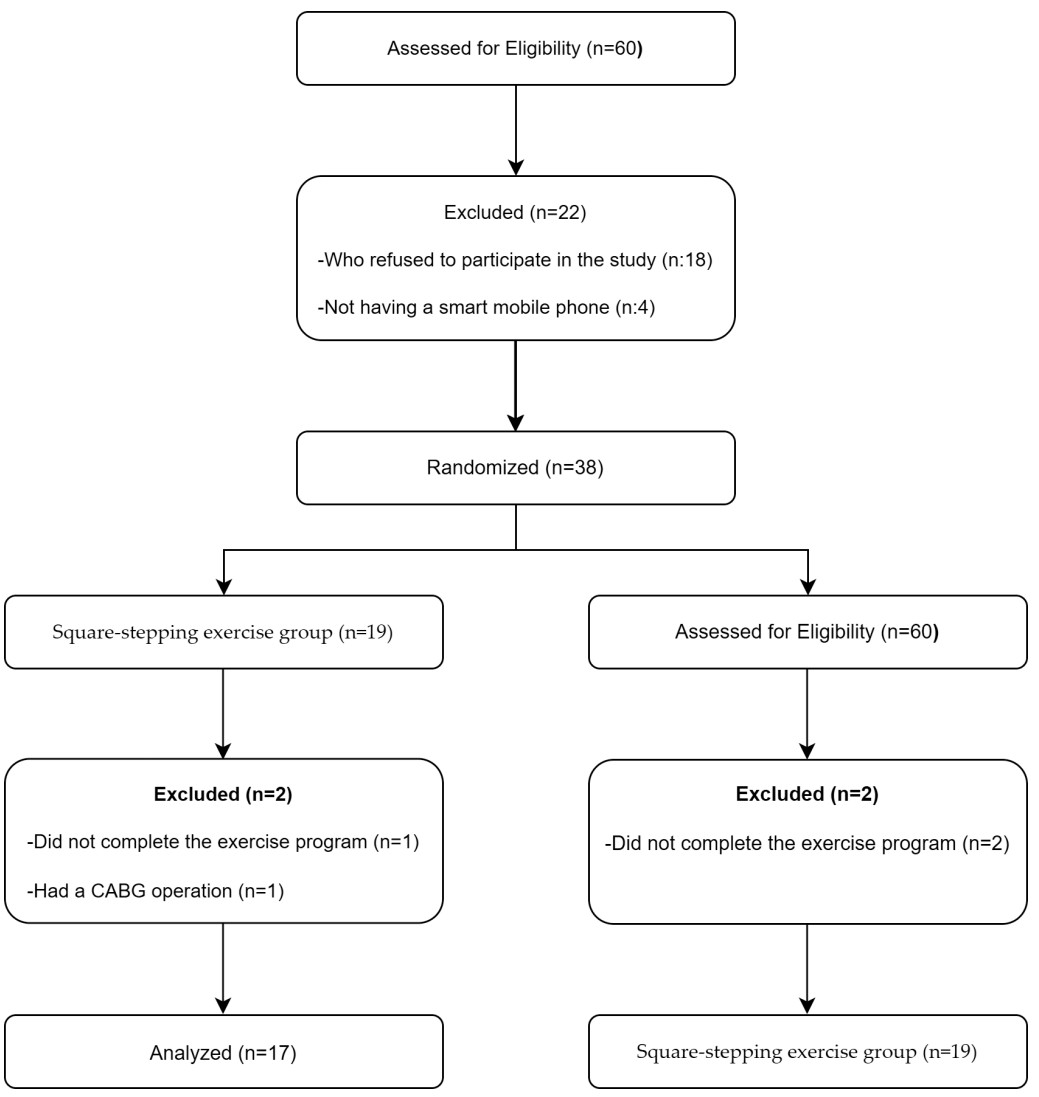

**Figure 1** **Flowchart illustrating the participant flow throughout the study.**

acute exacerbation for at least three weeks), $PaCO_2 \geq 70$ mmHg, conditions affecting cognitive function (including dementia, history of stroke, Parkinson's disease, Alzheimer's disease, major depression, schizophrenia, or other neurological and psychiatric disorders that could impair cognitive function), sensory impairments, severe chronic diseases (such as uncontrolled diabetes, severe cardiovascular diseases, chronic kidney failure, advanced liver disease, or any condition that could limit participation in physical activity and telerehabilitation), inability to walk, failure to follow the exercise program, and illiteracy. Written and informed consent was obtained for all participants. None of the participants participated in other experimental trials during the entire duration of the present study. The inclusion process of participants is illustrated in the flowchart shown in Fig. 1.

## Ethics approval

This study was approved by Bolu Abant Izzet Baysal University Clinical Researches Ethics Committee (Decision No:2020/14; Date: 04.02.2020) in line with the Declaration of Helsinki and registered with Clinical Trials (NCT04841005). Written informed consent was obtained from all participants prior to their involvement in the study, and the CONSORT guidelines were adhered to *Butcher et al. (2022)* and *Schulz, Altman & Moher (2010)*.

## Assessment tools

The primary outcome measure was cognitive function, assessed using the MoCA, which evaluates impairments across six cognitive domains with a total score ranging from 0 to 30, where higher scores indicate better cognitive performance (*Iamthanaporn, Wisitsartkul & Chuaychoo, 2023*). While the SMMSE was used as an inclusion criterion to exclude individuals with significant cognitive impairment, it primarily serves as a general cognitive screening tool. The SMMSE was used to screen for dementia, with a total score ranging from 0 to 30 across five cognitive domains. Lower scores indicate greater cognitive impairment, with a commonly used cutoff of 23 to identify possible dementia (*Molloy & Standish, 1997*). Since SMMSE is less sensitive to mild cognitive changes, MoCA was chosen as the primary assessment tool to detect potential cognitive improvements following the intervention. The Biodex Balance System (Biodex, New York, NY, USA) assessed balance with sway index measurements under four conditions: eyes open/closed on a firm and foam surface, where lower scores indicate better stability (*Kaya et al., 2023*). The mMRC scale assessed dyspnea on a 0–4 scale, with four indicating severe breathlessness (*Lewthwaite, Jensen & Ekström, 2021*). The CAT evaluated the impact of symptoms on daily life, with each of the eight items scored from 0 (no impact) to 5 (severe impact). Higher total scores indicate greater symptom burden and worse health status (*Tomaszewski et al., 2023*). Cough, sputum production, and fatigue were recorded based on self-reported symptoms and evaluated in a binary format as either present or absent. The modified CCI assessed comorbidity severity, with higher scores indicating more severe conditions (*Charlson et al., 2022*). Pulmonary function was assessed using the Minispir Spirometer (Medica International Research, Rome, Italy) in accordance with ATS/ERS standards. Participants performed three acceptable and reproducible maneuvers, and the best value for each parameter ($FEV_1$, FVC, $FEV_1$/FVC ratio, $FEF_{25–75}$, and PEF) was recorded to ensure measurement accuracy (*Stanojevic et al., 2022*). All assessments were conducted by trained physiotherapists with standardized instructions in accordance with the official guidelines for each tool. Data were collected during face-to-face evaluations at baseline and after 8 weeks and immediately recorded in anonymized forms.

## Exercise protocols

All sessions were delivered in real time *via* supervised telerehabilitation using the WhatsApp video call platform, which enabled continuous visual and verbal interaction between the physiotherapist and the participant throughout the exercise session. Each session included diaphragmatic breathing and thoracic expansion exercises (5 min), warm-ups (10 min), and cool-downs (10 min), ensuring safety and adherence to the intervention.

## Square-stepping exercise protocol

The SSE group performed SSE three times per week for eight weeks, with each session consisting of 10 min of warm-up exercises, 30 min of SSE training, and 10 min of cool-down exercises. The SSE program followed a progressive structure, starting with basic step patterns and increasing in complexity over time. The exercise was conducted on a 100 × 250 cm mat, divided into 40 squares (25 × 25 cm each) (*Siqueira, Shigematsu & Sebastião, 2024*). Participants performed forward, backward, sideways, and diagonal steps, progressing through 196 step patterns categorized into eight levels (Easy 1–2, Intermediate 3–5, Advanced 6–8), with each pattern consisting of 2 to 16 steps depending on its difficulty (*Siqueira, Shigematsu & Sebastião, 2024*).

Each step pattern was repeated 4 to 10 times, with participants moving across the mat and returning at a normal walking pace. While no fixed stepping cadence was set, participants were expected to complete each pattern within 15–20 s. Progression was structured as follows: easy patterns in weeks 1–2, intermediate patterns in weeks 3–4, intermediate to advanced patterns in weeks 5–6, and advanced patterns in the final two weeks. If participants failed to meet the timing or made frequent stepping errors, the progression was paused, and reinforcement at the current level continued until accuracy was achieved. Exercises were performed at a self-selected pace resembling normal walking speed, with visual demonstration provided by the physiotherapist prior to each new pattern.

## Strengthening exercise protocol

The SE group participated in resistance training sessions three times weekly for eight weeks. Each session included a warm-up phase (10 min), a main strengthening component, and a cool-down phase (10 min). Warm-up exercises consisted of light aerobic movements and dynamic stretching, while cool-down activities included breathing exercises and static stretching to promote recovery and reduce post-exercise discomfort. Each session included three sets per exercise, with 8 to 12 repetitions per set and 2–3 min rest intervals between sets. Elastic resistance bands were used to provide progressive loading, in accordance with the American College of Sports Medicine (ACSM) guideline (*Garvey et al., 2016*). Exercise intensity was regulated using the Borg Rating of Perceived Exertion (RPE) scale, with a target range of 12 to 14, corresponding to a perception of "somewhat hard" effort. Resistance levels were progressively increased when participants could perform more than fifteen repetitions easily for two consecutive sessions.

The SE group performed various exercises targeting muscle strength, including shoulder abduction, elbow flexion, and shoulder press movements. Additional exercises included horizontal abduction, punching exercises with an elastic band, triceps strengthening, shoulder external rotation, hip abduction with external rotation, and a standing exercise from a seated position. Strength training followed a progressive overload principle, starting with eight repetitions at the initial intensity. As participants were able to perform twelve repetitions easily, the number of repetitions was gradually increased.

A 2–3 min rest period was provided between sets. When participants could perform more than fifteen repetitions easily for two consecutive sessions without severe muscle or joint pain, the elastic band color was changed to increase resistance. After the resistance was

adjusted, participants restarted at eight repetitions with the new intensity and continued progressing accordingly.

## Sample size

The primary outcome measure was cognitive function, assessed by the MoCA. Power analysis was conducted using G*Power software (version 3.1.9.7) to determine the required sample size for the study. *Sanders et al. (2019)* reported an effect size of $d = 1.02$ for older adults engaging in multicomponent exercises, supporting the feasibility of this sample size. Based on a significance level ($\alpha = 0.05$) and a power of 80% ($1 - \beta = 0.80$), the power analysis determined that a minimum of 13 participants per group was required. To account for potential dropouts, 19 participants were initially recruited for each group, and the study was successfully completed with 17 participants in each group.

## Statistical analysis

Data analysis was conducted using IBM SPSS version 25.0 (IBM, Armonk, NY, USA). The Shapiro–Wilk test assessed data normality. Descriptive statistics, including median values, interquartile ranges, numbers, and percentages, were calculated based on variable types. Parametric tests were used for normally distributed data, while nonparametric tests were applied for non-normally distributed data.

A per protocol analysis was performed, meaning that only participants who completed the intervention were included in the final analysis. While this approach ensures that the intervention's effects are evaluated under optimal conditions, it does not account for participants who discontinued the program. Future studies may consider using an intention-to-treat (ITT) analysis to enhance the generalizability of the findings.

For group comparisons, the independent sample $t$-test was used for parametric data, and the Mann–Whitney $U$ test for non-parametric data. Differences between measurements were analyzed using a paired sample $t$-test for parametric data and the Wilcoxon signed-rank test for nonparametric data. A $p$-value of $< 0.05$ was considered statistically significant.

## RESULTS

A total of 60 candidates were evaluated, with 38 meeting the inclusion criteria and completing the study. The demographic characteristics of the participants are presented in Table 1. There were no significant differences between the groups in terms of age ($p = 0.324$), body mass index (BMI) ($p = 0.717$), mMRC ($p = 0.078$), CAT ($p = 0.375$), SMMSE ($p = 0.274$), modified CCI ($p = 0.182$), FEV$_1$ ($p = 0.485$), and FEV$_1$/FVC ($p = 0.274$).

Table 2 presents the comparison of MoCA scores between the SSE and SE groups at baseline, post-intervention, and between groups. No significant differences were found between the groups after the intervention ($p = 0.927$), however, both the SSE ($p = 0.001$) and SE ($p = 0.001$) groups demonstrated significant improvements in MoCA scores after 8 weeks. These results indicate that both types of exercise interventions had comparable effects on cognitive functions. Initially, it might have been expected that SSE would lead

**Table 1** Baseline demographic and clinical characteristics of the square-stepping exercise (SSE) group and strengthening exercise (SE) group.

| | SSE group ($n = 17$) | | SE group ($n = 17$) | | |
|---|---|---|---|---|---|
| | Mean ± SD | min–max | Mean ± SD | min–max | Between group p |
| Age (year) | 62.71 ± 4.74 | 55–73 | 65.12 ± 8.66 | 50–79 | 0.324 ($t = -1.007$) |
| Height (cm) | 169.12 ± 5.19 | 160–177 | 169.18 ± 5.4 | 155–175 | 0.973 ($z = -0.053$) |
| Weight (kg) | 76.47 ± 11.39 | 54–94 | 78.06 ± 16.13 | 52–118 | 0.742 ($t = -0.332$) |
| BMI (kg/m$^2$) | 26.69 ± 3.55 | 19.36–32.53 | 27.27 ± 5.51 | 17.99–40.83 | 0.717 ($t = -0.365$) |
| Smoking exposure (pack-years) | 38.26 ± 19.55 | 0–80 | 37.94 ± 18.46 | 0–70 | 0.961 ($t = 0.05$) |
| Exacerbations (n) | 0.41 ± 0.87 | 0–3 | 0.29 ± 0.69 | 0–2 | 0.786 ($z = -0.415$) |
| Disease duration (year) | 14.82 ± 8.31 | 2–27 | 15.93 ± 8.29 | 0.75–30 | 0.701 ($t = -0.387$) |
| mMRC (0–4) | 1.71 ± 1.16 | 0–4 | 1.82 ± 1.01 | 0–4 | 0.078 ($z = -0.396$) |
| CAT (0–40) | 18.06 ± 6.52 | 7–27 | 20.35 ± 4.43 | 13–25 | 0.375 ($z = -0.917$) |
| SMMSE (0–30) | 24.94 ± 2.22 | 23–29 | 25.82 ± 2.53 | 23–30 | 0.274 ($z = -1.147$) |
| Modified CCI (0–37) | 3 ± 1.73 | 1–8 | 3.35 ± 1.11 | 2–6 | 0.182 ($z = -1.417$) |
| FVC (%) | 82.18 ± 22.55 | 53–126 | 75.24 ± 23.83 | 35–111 | 0.39 ($t = 0.872$) |
| FEV$_1$ (%) | 64.24 ± 20.1 | 38–121 | 59 ± 23 | 32–108 | 0.485 ($t = 0.707$) |
| FEV$_1$/FVC | 63 ± 10.09 | 39–70 | 61.53 ± 9.91 | 40–70 | 0.274 ($z = -1.123$) |
| FEF$_{25-75}$ (%) | 30.82 ± 13.26 | 14–59 | 28.88 ± 11.87 | 15–51 | 0.656 ($t = 0.45$) |
| PEF (%) | 57.59 ± 17.79 | 30–93 | 53.18 ± 22.04 | 30–91 | 0.496 ($z = -0.69$) |
| Active smoker, n (%) | 10 (58.8) | | 10 (58.8) | | |
| Dyspnea, n (%), Yes | 10 (%58.82) | | 8 (%47.06) | | |
| Cough, n (%), Yes | 10 (%58.82) | | 9 (%52.94) | | |
| Sputum, n (%), Yes | 13 (%76.47) | | 12 (%70.59) | | |
| Fatigue, n (%), Yes | 16 (%94.12) | | 15 (%88.24) | | |

Notes.

$p < 0.05$.

$t$, $t$ test in independent groups; $z$, Mann–Whitney $U$ test; SD, Standard deviation; Med, Median; min–max, Minimum–maximum; BMI, Body mass index; mMRC, Modified Medical Research Council Dyspnea Scale; CAT, COPD assessment test; SMMSE, Standardized Mini-Mental State Examination; CCI, Charlson comorbidity index; FVC, Forced vital capacity; FEV$_1$, Forced expiratory volume in the first second; FEF$_{25-75}$, Forced expiratory flow mid expiratory phase; PEF, Peak expiratory flow rate.

**Table 2** Pre and post-intervention comparison of Montreal Cognitive Assessment (MoCA) scores in the square-stepping exercise (SSE) and strengthening exercise (SE) groups.

| | | SSE group ($n = 17$) | | SE group ($n = 17$) | | |
|---|---|---|---|---|---|---|
| | | Mean ± SD | min–max | Mean ± SD | min–max | Between group p |
| MoCA (19–25) | Baseline | 21.65 ± 1.69 | 19–25 | 22.24 ± 1.35 | 20–25 | |
| | Post intervention | 24.35 ± 2 | 21–27 | 24.29 ± 1.69 | 21–27 | 0.927 ($t = 0.093$) |
| | Group differences p | 0.001$^*$ (t = $-7.948$) | | 0.001$^*$ (t = $-4.757$) | | |

Notes.

$^*p < 0.05$.

$t$, $t$ test in independent groups; SD, Standard deviation; Med, Median; min–max, Minimum–maximum.

to greater improvements in cognitive functions; however, the findings suggest that both interventions are similarly effective in enhancing cognitive outcomes in individuals with COPD. This underscores the need for further research to explore the comparative effects of these interventions in more detail.

Table 3 compares balance variables, assessed with the Biodex Balance System, between the SSE and SE groups at baseline, post-intervention, and between groups. Post-intervention, no significant differences were observed between the groups for the medial/lateral stability index ($p = 0.085$), eyes open on a firm surface ($p = 0.192$), eyes closed on a firm surface ($p = 0.732$), eyes open on a foam surface ($p = 0.193$), or eyes closed on a foam surface ($p = 0.058$). The SSE group showed a significant decrease in the overall stability index from $1.17 \pm 0.32$ to $0.95 \pm 0.31$ ($p = 0.001$), and in the anterior/posterior index from $1.00 \pm 0.33$ to $0.78 \pm 0.30$ ($p = 0.001$). These represent approximately 18.8% and 22.0% relative improvements, respectively. In contrast, the SE group did not show statistically significant changes in these parameters. These changes in sway indices measured by the Biodex Balance System indicate improved postural control and reduced fall risk, particularly for dynamic balance. This suggests that both interventions were effective in supporting cognitive function, with SSE demonstrating an added benefit in balance-related outcomes.

However, significant differences were observed in the overall stability index ($p = 0.014$) and anterior/posterior stability index ($p = 0.05$) between the SSE and SE groups after the intervention. The SSE group showed significant improvements in the overall stability index ($p = 0.001$), anterior/posterior stability index ($p = 0.001$), medial/lateral stability index ($p = 0.001$), eyes closed on a firm surface ($p = 0.001$), eyes open on a foam surface ($p = 0.001$), and eyes closed on a foam surface ($p = 0.004$). In contrast, the SE group showed significant improvement only in the eyes open on a firm surface condition ($p = 0.029$) after the intervention.

In the SSE group, most participants showed concurrent improvements in both cognitive and balance domains. One participant, for example, increased their MoCA score from 23 to 30 ($+7$ points) and improved their overall stability index from 1.5 to 0.7 ($-0.8$).

Conversely, some participants exhibited limited response. In the SE group, a participant showed no change in MoCA score (27 to 27) and no improvement in overall stability index (1.2 to 1.2), despite attending all sessions. Such examples indicate the range of responsiveness across individuals and help contextualize the group-level findings.

## DISCUSSION

This randomized clinical trial aimed to compare the effects of SSE and SE on cognitive function and balance in individuals with COPD when delivered through telerehabilitation. While no significant differences were found between the groups regarding cognitive function improvements, both groups showed significant progress post-intervention. These results align with existing literature highlighting the positive effects of physical activity on cognitive functions in individuals with COPD (*Sanders et al., 2019*). The results obtained are consistent with existing literature highlighting the positive effects of physical activity on cognitive functions in individuals with COPD. Previous studies have shown that regular physical exercise supports cognitive functions and may even slow cognitive decline. In this context, a study conducted by *Ding et al. (2025)* reported that aerobic exercise programs in patients with mild to moderate COPD led to significant improvements in

**Table 3 Cognitive performance and postural balance outcomes before and after the intervention in the Square-Stepping Exercise (SSE) and Strengthening Exercise (SE) groups.** Measures include MoCA (Montreal Cognitive Assessment), Overall Stability Index, and Anterio.

| | | SSE group ($n = 17$) | | | SE group ($n = 17$) | | | |
|---|---|---|---|---|---|---|---|---|
| | | Mean ± SD | Med (IQR) | min–max | Mean ± SD | Med (IQR) | min–max | Between group p |
| Overall stability index | Baseline | 1.17 ± 0.32 | 1.1 (0.9–1.4) | 0.8–1.7 | 1.29 ± 0.38 | 1.3 (1.1–1.5) | 0.6–1.9 | |
| | Post intervention | 0.95 ± 0.31 | 0.9 (0.7–1.15) | 0.3–1.6 | 1.24 ± 0.33 | 1.3 (1–1.4) | 0.7–1.9 | 0.014* (t = −2.601) |
| | Group Differences p | 0.001* (z = −3.37) | | | 0.068 (z = −1.826) | | | |
| Anterior/ Posterior stability index | Baseline | 1 ± 0.33 | 0.9 (0.75–1.25) | 0.5–1.6 | 0.97 ± 0.35 | 1 (0.8–1.1) | 0.4–1.7 | |
| | Post intervention | 0.78 ± 0.3 | 0.8 (0.55–1) | 0.2–1.2 | 0.97 ± 0.26 | 1 (0.8–1.2) | 0.4–1.3 | 0.05* (t = −2.039) |
| | Group Differences p | 0.001* (t = 3.997) | | | 0.822 (z = −0.225) | | | |
| Medial/ Lateral stability index | Baseline | 0.79 ± 0.28 | 0.8 (0.65–1.05) | 0.2–1.2 | 0.66 ± 0.32 | 0.7 (0.4–0.9) | 0.2–1.3 | |
| | Post intervention | 0.46 ± 0.25 | 0.4 (0.2–0.65) | 0.2–1 | 0.62 ± 0.28 | 0.7 (0.35–0.85) | 0.3–1.1 | 0.085 (z = −1.737) |
| | Group Differences p | 0.001* (t = 5.256) | | | 0.083 (z = −1.732) | | | |
| Eyes open hard surface | Baseline | 0.68 ± 0.29 | 0.6 (0.44–0.95) | 0.28–1.2 | 0.64 ± 0.28 | 0.56 (0.47–0.72) | 0.37–1.59 | |
| | Post intervention | 0.61 ± 0.24 | 0.56 (0.45–0.69) | 0.31–1.12 | 0.73 ± 0.28 | 0.69 (0.51–0.91) | 0.36–1.44 | 0.192 (t = −1.334) |
| | Group Differences p | 0.794 (z = −0.261) | | | 0.029* (t = −2.399) | | | |
| Eyes closed hard surface | Baseline | 1.21 ± 0.38 | 1.19 (0.87–1.54) | 0.54–1.89 | 1.06 ± 0.42 | 1.06 (0.77–1.35) | 0.39–1.97 | |
| | Post intervention | 1.01 ± 0.34 | 1.04 (0.73–1.28) | 0.42–1.63 | 1.05 ± 0.36 | 1.14 (0.79–1.3) | 0.47–1.68 | 0.732 (t = −0.345) |
| | Group Differences p | 0.001* (t = 7.576) | | | 0.586 (z = −0.545) | | | |
| Eyes open foam surface | Baseline | 1.22 ± 0.35 | 1.19 (0.97–1.51) | 0.55-1.84 | 1.22 ± 0.38 | 1.14 (1.04–1.32) | 0.66–2.09 | |
| | Post intervention | 0.99 ± 0.32 | 0.89 (0.73–1.25) | 0.53–1.57 | 1.14 ± 0.33 | 1.07 (0.84–1.35) | 0.61–1.81 | 0.193 (t = −1.33) |
| | Group Differences p | 0.001* (z = −3.481) | | | 0.089 (t = 1.81) | | | |
| Eyes closed foam surface | Baseline | 3.05 ± 0.77 | 2.96 (2.46–3.77) | 1.91–4.25 | 3.22 ± 0.6 | 3.17 (2.84–3.6) | 1.99–4.34 | |
| | Post intervention | 2.77 ± 0.68 | 2.57 (2.2–3.43) | 1.77–4.05 | 3.2 ± 0.6 | 3.09 (2.8–3.82) | 1.95–4.07 | 0.058 (t = −1.966) |
| | Group Differences p | 0.004* (z = −2.914) | | | 0.768 (t = 0.3) | | | |

**Notes.**
*$p < 0.05$.

For intergroup analyses $t$, $t$ test in independent groups; $z$, Mann–Whitney $U$ test; for in-group analyses $t$, $t$ test in dependent groups; $z$, Wilcoxon paired two-sample test; SD, Standard deviation; Med, Median; IQR, Interquartile range.

cognitive functions (*Ding et al., 2025*). Similarly, another study reported that various types of exercises, especially multitasking exercises, enhanced cognitive flexibility and problem-solving skills in these patients (*Ding et al., 2024*).

While the improvement in MoCA scores observed in this study was statistically significant, statistical significance does not necessarily imply clinical relevance. To determine whether the observed changes indicate true cognitive improvement rather than measurement variability, the concept of minimal clinically important difference (MCID) should be considered. In populations with chronic illness or cognitive impairment, a 2-point change in MoCA scores has been proposed as a clinically meaningful threshold (*France et al., 2021*). In our study, the mean improvement in the SSE group exceeded

this threshold, suggesting that the cognitive gains may be both statistically and clinically relevant.

Nonetheless, previous literature has emphasized that changes in MoCA scores should be interpreted with caution, especially in studies with small sample sizes (*Lindvall et al., 2024*). Given the limited sample in our trial, future research incorporating anchor-based analyses or population-specific MCID estimates is needed to confirm the clinical significance of these findings. Cognitive impairment is common among individuals with COPD, yet intervention-based studies targeting its improvement remain scarce. While the cognitive benefits of physical exercise are well-documented in older adults, our findings indicate that similar improvements can also be achieved in individuals with chronic conditions such as COPD (*Boa Sorte Silva et al., 2018*). Although both groups showed statistically significant within-group improvements in MoCA scores, no significant between-group differences were observed. Several factors may explain this finding. First, participants in both groups had relatively high baseline MoCA scores, which may have resulted in a ceiling effect limiting the potential for measurable improvement. Second, the cognitive demands of the SSE protocol, while structured, may not have been sufficiently complex to elicit larger cognitive gains compared to general strengthening exercises. Third, the intervention period of eight weeks may have been too short to observe significant between-group divergence in cognitive outcomes. Future studies with longer durations, more challenging cognitive-motor tasks, and baseline stratification by cognitive status may help clarify these effects.

Exercises combining physical and cognitive tasks have been shown to be more effective for enhancing cognitive skills in older adults compared to single-task exercises (*Law et al., 2014*). The cognitive improvements observed in the SSE group suggest that multitasking exercises may offer greater benefits in this area. As SSE integrates both physical effort and cognitive challenges, it likely creates a synergistic effect, enhancing both mental and physical functions simultaneously.

Both exercise regimes showed significant improvements in cognitive functions, with the SSE group exhibiting more substantial enhancements. However, the relationship between cognitive impairment and physical activity levels is crucial in individuals with COPD. Cognitive decline can lead to reduced levels of physical activity, creating a detrimental cycle. Although we did not directly assess the impact of cognitive improvements on quality of life in this study, literature suggests that enhancing cognitive function through physical exercises like SE and SSE may contribute to increased physical activity levels, which in turn can improve overall quality of life. Strengthening exercises, by improving muscle strength and postural stability, may indirectly enhance functional capacity and thereby lead to improved quality of life by enabling greater physical activity. In this context, future studies should consider evaluating the effects of these interventions on quality of life to provide a more comprehensive understanding of their benefits in COPD management.

Studies have highlighted the positive effects of multitasking exercises on cognitive functions (*Sok et al., 2021*; *Teixeira et al., 2013*). *Teixeira et al. (2013)* reported significant cognitive improvements in older adults following SSE. Another study during the COVID-19 pandemic examined the short-term effects of home-based online SSE on cognitive and social

functions in inactive older adults (*Gan et al., 2022*). In comparison to the previous study, our study observed cognitive function improvements in both groups of individuals with COPD, with significant balance improvements noted particularly in the SSE group. Both studies highlight the potential of multitasking exercises to enhance cognitive abilities, such as executive functions. Additionally, these exercises, which can be effectively implemented online, are crucial for maintaining physical activity, especially during periods of social isolation, such as the COVID-19 pandemic (*Sturnieks, St George & Lord, 2008*). These results parallel the findings of our study and suggest that multitasking exercises can be especially beneficial for older adults and individuals with chronic diseases.

The SSE group demonstrated significant improvements in the overall stability index and anterior/posterior stability index compared to the SE group. To elucidate the mechanism behind SSE's advantage in these indicators, it's important to consider that SSE is a multifaceted exercise program that integrates physical movements with cognitive challenges. Such exercises require participants to be both physically and mentally active. Specifically, SSE challenges participants to step in various directions and remember complex step patterns, enhancing the coordination between the brain and muscles, particularly improving neuromuscular control over balance.

The effective role of SSE in balance is facilitated not only through direct impacts on muscle strength and postural stability but also by enhancing proprioceptive sensations and optimizing motor skills. This process can significantly reduce the risk of falls, especially in elderly individuals or those with chronic conditions. Regular practice of SSE significantly contributes to improvements in anterior and posterior balance indices because these exercises enhance an individual's ability to maintain balance, enabling more stable and controlled movements, which explains the improvements in the overall stability index. Improvements in balance parameters, particularly in the SSE group, are not only statistically significant but also clinically meaningful. The observed improvements in balance performance, particularly in the SSE group, have important clinical implications. Reductions in the overall and anterior/posterior stability indices suggest enhanced postural control, which is closely linked to fall risk in individuals with COPD. Previous research has shown that impaired balance is a strong predictor of falls and subsequent disability in this population (*Liwsrisakun et al., 2019*; *Hao et al., 2024*). Therefore, interventions such as SSE, which incorporate multidirectional movement and cognitive-motor integration, may contribute not only to postural stability but also to fall prevention. These benefits are particularly valuable in COPD rehabilitation, where fall-related injuries can severely impact quality of life and functional independence. The Biodex stability indices are validated markers for fall risk in older and chronically ill populations. The observed gains may be attributed to the multitasking nature of SSE, which engages executive functions and sensorimotor coordination simultaneously. This is supported by previous studies that identified dual-task balance training as superior to traditional strength training in improving dynamic stability (*Beauchamp et al., 2012*; *Shigematsu & Okura, 2006*). These results indicate that exercises addressing both cognitive and physical functions, such as square-stepping exercise, may help break the cycle of inactivity, cognitive impairment, and balance dysfunction commonly observed in individuals with COPD. Accordingly,

incorporating multitasking-based interventions into telerehabilitation programs could offer a valuable approach to improving overall functional capacity in this population.

Balance is a critical determinant of fall risk, especially in individuals with COPD (*Sturnieks, St George & Lord, 2008*). Research shows that balance impairments are prevalent in this population, with significant deficits in balance control compared to healthy individuals. A systematic review reported that individuals with COPD are four times more likely to experience falls than their healthy peers, emphasizing the need for thorough balance assessment and targeted interventions in this group (*Loughran et al., 2020*). The Berg Balance Scale and Timed Up and Go tests are commonly utilized to evaluate balance in individuals with COPD, although they may have limitations in fully capturing the extent of balance deficits (*Liwsrisakun et al., 2019*).

Muscle weakness is a well-established risk factor for falls and balance impairments in individuals with COPD. *Beauchamp et al. (2012)* highlighted the essential role of muscle strength in balance control, noting that deficits in peripheral muscle function are common in this population (*McLay, O'Hoski & Beauchamp, 2019*). Additionally, a sedentary lifestyle exacerbates these issues by impairing sensory integration and balance control (*Beauchamp et al., 2012*). Incorporating balance training into pulmonary rehabilitation programs has been proven to enhance balance performance and reduce fall risk, underscoring the need for targeted interventions in this population (*Law et al., 2014*). Balance-focused programs like SSE not only target balance but also cognitive skills, offering the potential to reduce the risk of falls in individuals with COPD (*Shigematsu & Okura, 2006*). The significant effect of SSE on balance may be due to the need for both physical and mental focus during the exercise. The observed improvements in postural control, particularly in the SSE group, have important clinical implications. Impaired balance is a known risk factor for falls in individuals with COPD, which can lead to hospitalization, reduced mobility, and lower quality of life. By targeting both motor and cognitive domains, SSE may offer a more comprehensive approach to fall prevention and functional independence in this population.

Participants had to focus on the squares and step sequences on the mat with each step, creating an exercise experience that actively engaged both balance and cognitive functions. This finding aligns with other multitasking exercises in the literature that specifically target balance (*Boa Sorte Silva et al., 2018*). Studies reporting the effectiveness of multitasking exercises in improving balance and postural control support our findings (*Shigematsu & Okura, 2006*). Additionally, it has been shown that such exercises can be a feasible and effective option when implemented *via* telerehabilitation (*Law et al., 2014*).

The SE group also demonstrated improvements in some balance parameters post-intervention, though these were more limited compared to the SSE group. While SE may not directly target balance, increased muscle strength is known to enhance postural control and stability (*Kaya et al., 2023*). The SE group primarily performed exercises focused on building muscle strength, which led to indirect benefits for balance. However, as their program did not specifically target balance, their stability outcomes were under-standably less pronounced than those of the SSE group.

*Paneroni et al. (2015)* highlighted telerehabilitation as a safe and feasible option for COPD patients and emphasized the need for further research in this area. Participants continued their exercises at home *via* telerehabilitation after receiving face-to-face training, underscoring the value of home-based exercise programs. Our findings demonstrate that telerehabilitation can be effectively applied to both multitasking and strengthening exercise interventions. However, the feasibility of telerehabilitation may vary depending on patient characteristics. Age-related challenges such as limited digital literacy, sensory impairments, or discomfort with technology may negatively impact participation and adherence. While participants in this study showed high compliance, future research should investigate strategies to improve accessibility and address technological barriers, particularly for older adults or those with limited experience using digital tools.

However, this study has some limitations. Firstly, since the sample size was relatively small, the generalizability of the findings may be limited. Conducting studies with larger participant groups could further enrich the literature. Furthermore, as the study included only male participants, it remains unclear whether similar results would be observed in females. While this approach reduced biological variability within the sample, it also limits the generalizability of the findings to the broader COPD population. Future research should aim to include more diverse and representative samples to enhance external validity. In addition, the use of per-protocol analysis, while providing a clean estimate of intervention efficacy, excludes participants who did not complete the program and may overestimate effects. The lack of ITT analysis should be considered when interpreting the generalizability of our findings. Furthermore, while patient safety was ensured using pulse oximetry and the Borg scale for fatigue, the modified Borg scale for dyspnea could have provided a more detailed assessment of breathing difficulties during telerehabilitation. One of the limitations of this study is the lack of assessment regarding participants' educational level, socioeconomic status, and prior experience with digital technology. Since the intervention was delivered *via* telerehabilitation, these factors could have influenced the participants' ability to engage with the program effectively. Although the intervention period was sufficient to demonstrate short-term benefits, the long-term sustainability of these improvements remains unclear. Continued engagement in multitasking exercises may be required to maintain gains in cognitive and balance functions. Future research should include extended follow-up periods to assess the durability of effects and explore strategies that promote long-term adherence to telerehabilitation-based programs. Despite these limitations, the per-protocol analysis ensures that the reported results indicate the direct effects of the interventions on compliant participants, offering valuable insight into their clinical utility. Future studies should consider including these variables to better understand their impact on adherence and outcomes.

## CONCLUSIONS

This randomized study examined the effects of SSE and SE on cognitive functions and balance in individuals with COPD over an 8-week period. The results showed that both groups demonstrated significant improvements in cognitive function, with no statistically

significant differences between them. These findings are consistent with previous studies that emphasize the role of physical activity in supporting cognitive health in individuals with COPD.

Importantly, this study also revealed that SSE led to more pronounced improvements in dynamic balance compared to SE, as demonstrated by significant changes in objective metrics such as the overall stability index and anterior/posterior stability index, which was measured using the Biodex Balance System. These indices offer measurable insights into postural control and balance performance, reinforcing the clinical utility of SSE as a multitasking intervention.

These findings suggest that both interventions can be safely and effectively delivered *via* telerehabilitation, improving their accessibility, especially for individuals with limited mobility or access to in-person rehabilitation services. From a clinical perspective, the incorporation of multitasking exercises like SSE into pulmonary rehabilitation programs may not only support physical and cognitive outcomes but also contribute to fall prevention, which is a critical objective in COPD management.

While cognitive impairment is common in individuals with COPD, the number of intervention-based studies focusing on this aspect remains limited. Our findings contribute to filling this gap by demonstrating that structured physical activity, particularly with cognitively engaging components, may produce synergistic benefits. The necessity of such an approach is supported by the interplay between reduced physical activity, cognitive decline, and balance dysfunction, which together form a triad that exacerbates disability in this population.

Future research should investigate the long-term sustainability of the observed improvements and validate these findings across larger, more diverse populations. The use of additional neurocognitive and sensorimotor assessment tools may also help elucidate the mechanisms driving these outcomes. Overall, this study supports the integration of cognitive-motor training into routine pulmonary rehabilitation and lays the groundwork for the development of multidimensional, patient-centered intervention strategies in COPD management.

### Funding
The authors received no funding for this work.

### Competing Interests
The authors declare there are no competing interests.

### Author Contributions
- Alp Özel conceived and designed the experiments, performed the experiments, analyzed the data, prepared figures and/or tables, and approved the final draft.
- Eylem Tütün Yümin conceived and designed the experiments, performed the experiments, analyzed the data, authored or reviewed drafts of the article, and approved the final draft.

- Suat Konuk conceived and designed the experiments, performed the experiments, analyzed the data, authored or reviewed drafts of the article, and approved the final draft.

## Human Ethics

The following information was supplied relating to ethical approvals (*i.e.*, approving body and any reference numbers):

Bolu Abant Izzet Baysal University Clinical Researches Ethics Committee (Decision No: 2020/14; Date: 04.02.2020).

## Clinical Trial Ethics

The following information was supplied relating to ethical approvals (*i.e.*, approving body and any reference numbers):

This study received approval from the Bolu Abant İzzet Baysal University Clinical Researches Ethics Committee (Decision No: 2020/14; Date: 04.02.2020) and was registered with Clinical Trials (NCT04841005).

## Data Availability

The raw data are available in the Supplementary File.

## Clinical Trial Registration

The following information was supplied regarding Clinical Trial registration:

NCT04841005.

## Supplemental Information

Supplemental information for this article can be found online at http://dx.doi.org/10.7717/peerj.19792#supplemental-information.

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
