# Peer review of "Comparative effects of square-stepping and strengthening exercises on cognitive and balance functions in chronic obstructive pulmonary disease: a randomized clinical trial"

_PeerJ, doi:10.7717/peerj.19792_

## Round 0.1 · original submission · Major Revisions

Reviewer 1 ·

Basic reporting

The work is somehow new, but several limitations, from the English to the technical, hinder grasping the main point of the work.

1. The abstract must be re-written, focusing on the technical aspects of the proposed model, the main experimental results, and the metrics used in the evaluation. Briefly discuss how the proposed model is superior.
2. Additionally, method names should not be capitalized. Moreover, it is not the best practice to employ abbreviations in the abstract, they should be used when the term is introduced for the first time.
3. The contribution of the current study must be briefly discussed as bullet points in the introduction. And motivation must also be discussed in the manuscript.
4. The overall organization of the manuscript is not discussed anywhere in the manuscript. Please add the same in the introduction section of the manuscript.
5. By considering the current form of the conclusion section, it is hard to understand by the Journal readers. It should be extended with new sentences about the necessity and contributions of the study by considering the authors' opinions about the experimental results derived from some other well-known objective evaluation values if possible.
6. English proofreading is strongly recommended for a better understanding of the study. Few sentences are written in passive voice and it is also observed that few sentences stopped abruptly.

Experimental design

1. Authors must discuss more about how the experimental study is performed.
2. Authors must discuss more about how the data is acquired and processed for evaluation.
3. Authors must also discuss about the automation approaches for data evaluation and experimentation.
4. Please add more adequate information about COPD classification.
5. Please provide more details about the Exercise Protocols in technical grounds to reproduce the results.

Validity of the findings

More details and observations must be published to make the study evident and also discuss about the grounds facts associated with the clinical trail data.

Additional comments

1. A methodological manuscript, like the current one, must formulate what problem needs to be solved and why existing techniques are insufficient.
2. The methods should be sufficiently accurately described or contain suitable references so that the manuscript reader has a fair possibility to reproduce the results. It should be clear which parts of the method are new and which are not.
3. The results should be presented with sufficient detail, including examples of successful and unsuccessful cases (if any).

Reviewer 2 ·

Basic reporting

The manuscript is clearly written. It is logically structured, supported by a comprehensive review of background literature, and includes well-labeled figures and tables. Referencing is also adequate and up to date. However, the current title is vague. I recommend: “Comparative Effects of Square-Stepping and Strengthening Exercises on Cognitive and Balance Functions in COPD: A Randomized Clinical Trial” This title better reflects and emphasizes the study's content. I also suggest improving the clarity and completeness of figure and table captions, particularly Table 3, to help readers more easily grasp the complexly presented results. Abbreviations such as SE, SSE, and MoCA should be clearly defined within the captions, not in the foot note. Please ensure consistency in terminology and provide definitions of each term to avoid confusion.

Experimental design

The use of a randomized controlled trial design, with telerehabilitation delivery, is timely and appropriate. Exercise interventions are well-described and based on existing guidelines (e.g., ACSM). However, only male participants were included. This is a notable limitation that should be explicitly stated in the introduction and emphasized in the limitations section. Although MoCA is an appropriate tool, its minimal clinically important difference (MCID) should be discussed to contextualize whether observed changes are meaningful. Additionally, the study used per-protocol analysis but not ITT. This limits generalizability, and I recommend acknowledging this limitation more clearly.

Validity of the findings

Statistically significant within-group improvements are clearly reported. The study finds greater improvements in balance (but not cognitive function) for SSE compared to SE. However, the discussion focuses on p-values without interpreting how observed differences affect function (e.g., fall risk). Adding clinical context would strengthen conclusions. Additionally, the lack of between-group differences in MoCA is under-discussed. Possible reasons (e.g., ceiling effects, task complexity, short duration, etc) should be considered.

Additional comments

This study is well-conducted and contributes to the field by testing telerehabilitation-based multitasking exercises in a COPD population. It is a novel and practical approach. However, several areas can be improved, for example: clearly define the clinical implications of the findings, discuss feasibility and barriers to telerehabilitation adherence, especially considering participant age, explicitly state that results are limited to male participants, and possibly expand on the long-term applicability of the interventions.

---

## Round 0.2 · accepted · Accept

The manuscript has been improved considerably. It is ready to be published. Congratulations.

Reviewer 1 ·

Basic reporting

-

Experimental design

-

Validity of the findings

-

Additional comments

The authors have addressed all the recommendations of the reviewers in a reasonable manner; the manuscript, in its current form, may be considered for the further phase of the editorial process.

Reviewer 2 ·

Basic reporting

I appreciate the authors' careful and comprehensive revisions to the manuscript. The study addresses a relevant gap in COPD rehabilitation by comparing two exercise modalities through a telerehabilitation framework. The revised version demonstrates significant improvement in both reporting clarity and clinical contextualization. Below, I provide comments categorized by review sections:

1. Basic Reporting
• The revised title is now appropriately specific and better reflects the content and aims of the study. This strengthens the clarity and visibility of the article.
• Captions for figures and tables, particularly Table 3, have been improved with clearer definitions and units. Abbreviations (e.g., SSE, SE, MoCA) are now defined consistently within captions or text, reducing ambiguity.
• The overall manuscript is logically structured and well-written. The background provides a thorough review of relevant literature, and terminology is consistently used.

Experimental design

2. Experimental Design
• The rationale for including only male participants is now explicitly stated in both the Introduction and the Limitations section. While this approach minimizes variability, the authors correctly acknowledge its impact on generalizability.
• The manuscript now includes a discussion of the minimal clinically important difference (MCID) for the MoCA, which supports the clinical relevance of the cognitive outcomes.
• The choice of per-protocol analysis is justified, and the limitation of not including intention-to-treat (ITT) analysis is appropriately acknowledged.

Validity of the findings

3. Validity of Findings
• Statistical findings are clearly reported with correct use of within- and between-group comparisons. The manuscript now links balance improvements to clinically meaningful outcomes, particularly in relation to fall risk.
• The absence of between-group differences in MoCA scores is well addressed. The discussion now includes possible explanations (e.g., ceiling effect, task complexity, intervention duration), which is a valuable addition.
• The authors provide examples of individual-level variability, which help contextualize the group-level data.

Additional comments

4. General Comments
• The study contributes to the growing body of research supporting telerehabilitation approaches in chronic disease populations. Its focus on multitasking interventions is timely and practical.
• Clinical implications, including feasibility and accessibility of SSE for home-based delivery, are clearly described. The potential challenges for older adults regarding digital literacy are thoughtfully discussed.
• The discussion now better integrates the dual role of SSE in enhancing cognitive and balance outcomes. The long-term applicability of the intervention and recommendations for future research are well noted.